# Does Grounding Improve Radiology Report Generation? An Empirical Study on PadChest-GR

**Mohamed Aas-Alas** [ID]                                    MAASALA@PRHLT.UPV.ES
**Alberto Albiol** [ID]                                      ALALBIOL@PRHLT.UPV.ES
**Roberto Paredes** [ID]                                     RPAREDES@PRHLT.UPV.ES
*Campus de Vera, Universitat Politècnica València, Camí de Vera s/n, 46022 Valencia, Spain*

**Editors:** Accepted for publication at MIDL 2026

## Abstract

Radiology Report Generation (RRG) aims to automatically produce clinically accurate descriptions of medical images, yet current models often struggle with incomplete findings, generic phrasing, and hallucinations due to the absence of explicit grounding signals. To address these limitations, we propose a grounding-based RRG framework that integrates spatially localized visual evidence into the generation process. Our approach combines a vision encoder ViT with a language decoder LLM GPT-2 through a lightweight transformer-based bridging module inspired by Bridge-Enhanced Vision Encoder–Decoder (VED) architectures. Grounding is introduced using bounding boxes of anatomical regions and pathologies, enabling the model to attend to both global and localized features. We further adopt the region-to-text task, where the model generates findings directly from specific regions of interest. Experiments on the PadChest-GR dataset demonstrate that grounding substantially improves linguistic quality and clinical accuracy, with the full image plus grounding mask configuration achieving the strongest gains across BLEU, ROUGE-L, CIDEr, BERTScore, CheXbert F1, and RadGraph F1. Analyses also show that even partial or noisy grounding yields consistent benefits.

**Keywords:** Radiology Report Generation, Spatial Grounding, Region-to-Text Generation, Chest X-ray Interpretation.

## 1. Introduction

Radiology Report Generation (RRG) is a challenging task that aims to automatically produce detailed and clinically accurate textual descriptions of medical images, such as chest X-rays. This task requires not only recognizing visual abnormalities but also expressing them in a coherent and contextually appropriate medical narrative, similar to what a radiologist would write. Unlike traditional image captioning, RRG demands fine-grained understanding, reasoning, and domain-specific language generation.

However, RRG models face several problems because the decoder often implemented as a large language model (LLM) does not always know what to say or how to structure the report based solely on image features. Unlike Visual Question Answering (VQA), where a textual question provides a clear linguistic grounding and narrows the output space, RRG lacks an explicit guiding signal that constrains or directs the generation process. As a result, generated reports may be incomplete, overly generic, or contain hallucinated findings.

To address these challenges, we present a systematic study of an RRG model that incorporates grounding information to guide the generation process. Our primary motivation

is to rigorously assess the real impact of grounding by bounding boxes corresponding to detected anatomical regions and pathologies. Through ablation studies, we analyze whether and to what extent such grounding improves the factual accuracy, clinical relevance, and descriptive quality of the generated findings.

To achieve this, we first introduce a lightweight transformer-based bridging module that connects the visual encoder and language decoder, effectively integrating grounding information into the report generation process without significantly increasing computational cost. This component enables the model to attend to localized visual cues such as anatomical structures and pathological regions while maintaining global contextual awareness. Secondly, we propose a novel grounding method that leverages bounding boxes corresponding to detected anatomical regions and pathologies to explicitly guide the decoder toward relevant visual evidence during text generation. This approach also adopts the Region-to-Text task, in which the model generates a grounded finding directly from a specific region of interest.

For reproducibility and to facilitate further research in grounded Radiology Report Generation, a complete implementation of our approach is provided in a public repository, with all source code on GitHub[1]. We aim to promote transparency and foster future advancements by making our grounding-based RRG framework easily accessible to the research community.

## 2. Related work

Recent progress in multimodal models has centered on model architectures for vision–language fusion and strategies for spatial grounding. Advances in these areas have enabled finer alignment between visual features and textual reasoning. In radiology, models capable of generating bounding boxes or spatial cues represent a crucial step toward grounded visual understanding. The task we address, region-to-text generation, differs in that it uses the extracted region features (e.g., bounding boxes) to generate grounded findings rather than predicting regions themselves. Nevertheless, these works remain highly relevant as they demonstrate effective methods for integrating spatial grounding with clinical reasoning. RadVLM introduces multitask setups that couple instruction following with localization to steer radiology-specific behaviors (Deperrois et al., 2025). MAIRA-2 emphasizes grounded report generation by conditioning text on image evidence and explicitly linking descriptions to visual findings (Bannur et al., 2024). More recently, VividMed extends grounded radiology modeling by supporting versatile visual grounding with both bounding boxes and segmentation masks across 2D and 3D medical imaging modalities (Luo et al., 2025). Similarly, Knowledge to Sight explores how decomposing domain knowledge into visual attributes enhances abnormality grounding (Li et al., 2025), while NOVA extends these ideas to brain MRI, addressing rare anomaly localization and evaluation under distributional shift (Bercea et al., 2025).

Our focus aligns with region-to-text modeling in radiology, where the objective is to generate clinically grounded narratives directly from localized image features using bounding boxes. Related to this line of work, MAIRA-Seg (Sharma et al., 2025) generates radiology reports from chest X-rays while being explicitly aware of pixel-level anatomy, pathology,

---

1. https://github.com/LightVED-prhlt/RegionFormer

and medical devices through segmentation masks. However, such pixel-level annotations are substantially harder to obtain than bounding boxes, which limits scalability in real-world clinical settings. Additional related works that leverage bounding boxes include RoI-MedCap (Rubel et al., 2025a) and MoE-MSC (Rubel et al., 2025b), which encode the region of interest by directly integrating the bounding box into the image pixels using datasets such as MedTrinity-25M (Xie et al., 2025), our approach on the other hand treats bounding boxes as a separate input from the chest X-ray and does not visually embed them into the image. A more closely related paradigm is presented in Describe Anything in Medical Images (Xiao et al., 2025), which also treats regions separately from the image. While this work relies on datasets such as VinDr-CXR (Nguyen et al., 2022) that provide region-level labels, we instead use PadChest-GR (de Castro et al., 2025), a more recent dataset that includes bounding box–aligned free-text clinical findings. This enables supervised learning of region-level report generation with richer clinical descriptions.

Alternative prompting strategies have investigated how to elicit medically meaningful attention without requiring traditional box-level supervision. Visual Prompt Engineering for Vision–Language Models in Radiology demonstrates that structured prompts such as arrows or point-like cues can guide model attention and improve interpretability (Denner et al., 2025). While effective for classification, these strategies differ from region-to-text formulations where models must generate grounded findings from spatially localized visual evidence rather than classifying entire images.

Architecturally, a common design pattern is to bridge frozen or semi-frozen vision encoders with powerful language decoders through learnable interfaces. BLIP-2 exemplifies this approach by introducing a lightweight Q-Former to adapt image features for large language models, reducing training cost while preserving generalization (Li et al., 2023). ClipCap similarly maps CLIP image embeddings into a prefix for autoregressive decoders, forming a minimal yet effective conduit between modalities (Mokady et al., 2021). Scaling this idea, the Matryoshka Query Transformer introduces query-based designs that enable flexible, nested subsets of capabilities, facilitating efficient inference while retaining strong alignment (Hu et al., 2024). These bridges operationalize a broad trend: transformer-based backbones with compact, trainable adapters that translate vision features into token-like inputs consumable by language models.

With respect to grounding, region-aware VLMs vary in how they represent and inject spatial cues. Some encode regions as textual coordinates within the prompt, letting the language stream parse boxes directly; notable examples include Kosmos-2, Shikra, and PINK, which format box coordinates or referential tags inline with text to steer attention (Peng et al., 2023; Chen et al., 2023b; Xuan et al., 2024). Others insert region features as dedicated tokens derived from RoI-aligned crops, masked patches, points, or masks; GPT4RoI, Ferret, and MG-LLaVA exemplify this feature-level injection, fusing localized embeddings alongside global image tokens to enable referential comprehension and region-conditioned responses (Zhang et al., 2025; You et al., 2023; Zhao et al., 2024). A complementary line modifies the vision encoder to bias attention spatially, (Chen et al., 2023a) incorporates positional encodings or masking schemes that privilege specific areas without requiring explicit box tokens. GRILL pursues fine-grained alignment during pre-training by jointly learning to match textual spans with corresponding image regions, enforcing region–phrase correspondence that later benefits downstream grounding (Jin et al., 2023). Finally, interactive

systems such as RegionVLM encode user-provided points or scribbles as spatial trajectories, enabling iterative, fine-grained regional understanding without relying solely on bounding boxes and thus broadening the interface for clinical annotation and guidance (Lee et al., 2024).

Recent work (Delbrouck et al., 2025) has explored alternatives to free-form radiology report generation, notably Structured Radiology Report Generation (SRRG), which organizes reports into standardized, anatomy-based sections to improve consistency and evaluation reliability. However, SRRG mainly restructures text and does not explicitly ground each statement in visual evidence. In contrast, grounded RRG focuses on ensuring that generated language is directly supported by image-derived observations. Our work follows this grounded paradigm by constructing evidence-based question–answer pairs derived from radiographic findings, without enforcing rigid report templates. A combination of both approaches could be particularly promising, where SRRG provides structural guidance on where to focus within the image, while grounded generation ensures what is said remains visually faithful and clinically meaningful.

## 3. Methodology

Our framework establishes a unified pipeline for Grounded Radiology Report Generation (GRRG), combining visual encoding, spatial grounding, and language generation within a single end-to-end trainable model. Formally, given an input radiograph $I$ and its set of bounding boxes $\mathcal{B} = \{b_k\}_{k=1}^K$, the model aims to generate a coherent finding $\hat{Y} = (y_1, \ldots, y_T)$ conditioned on both global and localized visual cues.

### 3.1. Visual Encoding

Each image $I$ and its corresponding grounding mask $M$ are first projected into the visual embedding space through the vision encoder $\phi(\cdot)$, yielding a pooled global representation:

$$v_{\text{img}} = \phi(I), \qquad v_{\text{g.mask}} = \phi(M)$$

Depending on the configuration, either the image embedding $v_{\text{img}}$, the grounding mask embedding $v_{\text{g.mask}}$, or their concatenation is used:

$$v = \begin{cases} v_{\text{img}}, & \text{image-only mode} \\ v_{\text{g.mask}}, & \text{grounding mask-only mode} \\ [v_{\text{img}}; v_{\text{g.mask}}], & \text{image+grounding mask mode} \end{cases}$$

All embeddings are L2-normalized; in the concatenation case, the resulting vector $v$ is normalized after concatenation such that $\|v\|_2 = 1$.

### 3.2. Visual–Language Mapping

The visual representation $v \in \mathbb{R}^d$ corresponds to the vision encoder pooled image embedding (or the concatenation of image and grounding-mask embeddings, depending on the configuration). Although the vision encoder internally computes a sequence of patch-level

representations, in our implementation we explicitly use only the final pooled embedding, resulting in a single global visual embedding per input sample.

This visual representation is mapped to the large language model (LLM) embedding space through a learnable Transformer-based mapper $f_\theta$, which produces a fixed-length sequence of $L_p$ prefix tokens $P = [p_1, \ldots, p_{L_p}] \in \mathbb{R}^{L_p \times h}$, where $h$ is the LLM hidden size:

$$P = f_\theta(v) = \text{Transformer}\big([Wv]_{L_c} \,\|\, E_{\text{prefix}}\big)_{L_p}.$$

Here, $W \in \mathbb{R}^{h \times d}$ is a linear projection that expands the single visual embedding $v$ into a sequence of $L_c$ latent tokens, each compatible with the LLM embedding dimensionality. This expansion does not correspond to multiple image patches or spatial regions, but instead serves as a learned mechanism to unpack a compressed global visual representation into a sequence of token-like embeddings that can be processed by a Transformer.

The projected visual tokens are concatenated with a set of learnable prefix embeddings $E_{\text{prefix}}$, and the Transformer refines this joint representation through self-attention. The final $L_p$ tokens corresponding to the learned prefix positions form the output $P$, which is concatenated with the textual token embeddings and used as conditioning context for the LLM during generation (see Pseudocode in appendix A).

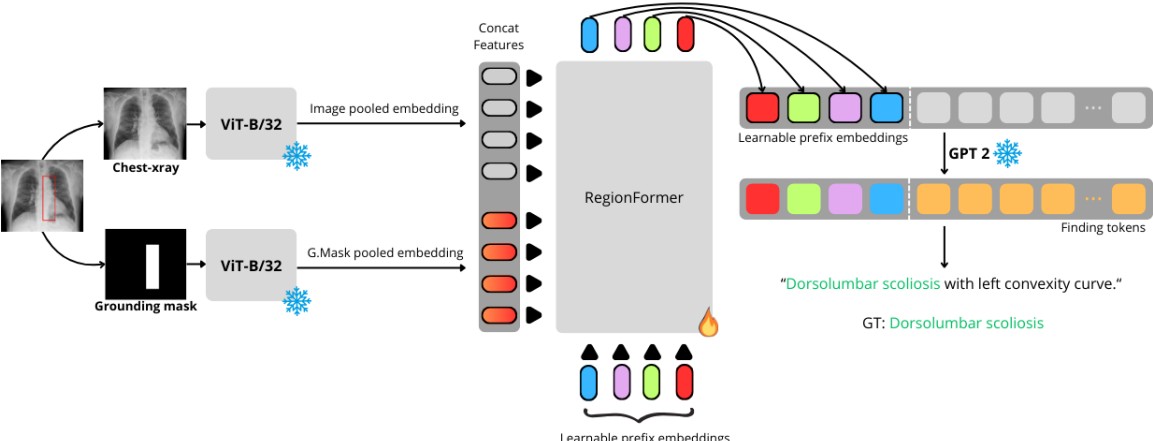

Figure 1: Our proposed architecture for grounded radiology report generation.

In conclusion, the RegionFormer module acts as the mapping network that bridges the gap between the visual and language components. Since the vision encoder and LLM operate in different embedding spaces, the visual features extracted by the vision encoder cannot be directly used by the LLM. RegionFormer addresses this by learning a transformation that maps the vision encoder image embeddings into the LLM's embedding space.

Each sample produces a visual embedding from the full image, the grounding mask or the concatenation of both. The visual embedding is processed by RegionFormer to generate a sequence of learnable prefix embeddings. Through self-attention, these prefix embeddings retrieve and encode the meaningful visual information contained in the visual embedding,

capturing both global context and grounded regional cues. They result in learned prefix embeddings which act as prefix tokens that are prepended to the textual input of the LLM, guiding the language model to generate radiology reports conditioned on both global context and spatially localized visual evidence.

### 3.3. Language Generation

Given token embeddings $E(y_1), \ldots, E(y_T)$, the input sequence to the large language model is constructed as:

$$Z = [P; E(y_1), \ldots, E(y_T)]$$

the LLM then predicts the probability of each next token conditioned on both the prefix and the previous tokens:

$$P(y_t \mid y_{<t}, P) = \mathrm{LLM}(Z)_t$$

The model is trained by minimizing the standard next-token cross-entropy loss:

$$\mathcal{L}_{\mathrm{CE}} = -\sum_{t=1}^{T} \log P(y_t \mid y_{<t}, P)$$

During inference, reports are generated autoregressively using beam search.

### 3.4. Grounded Generation Objective

Through the grounding mask embeddings, the model learns to associate textual tokens with localized image evidence. The overall generation function can be written as

$$\hat{Y} = \arg\max_Y P(Y \mid I, M) = \arg\max_Y P(Y \mid f_\theta([\phi(I); \phi(M)]))$$

thus ensuring that the generated finding $\hat{Y}$ is grounded both globally and regionally in the visual content.

Overall, our approach jointly models visual and textual modalities through learned prefix embeddings that bridge visual representations and autoregressive language generation, enabling spatially grounded and semantically coherent radiology report synthesis.

### 3.5. Implementation Details

Our proposed model for the Grounded Radiology Report Generation task utilizes a Bridge-Enhanced Vision Encoder Decoder (VED) architecture as shown in Figure 1. It integrates the VIT (Dosovitskiy et al., 2021) as a vision model and GPT-2 (Radford et al., 2019) as the language model, both of which remain frozen during training while a lightweight mapping network is learned.

#### 3.5.1. Vision Component

The Vision Transformer (ViT), more specifically the CLIP-based (Radford et al., 2021) ViT-B/32 model, is used as the vision encoder. Although architecturally similar to the standard ViT-B/32, the CLIP variant is trained from scratch on hundreds of millions of image–text

pairs, learning a visual embedding space that aligns naturally with linguistic semantics. This makes CLIP-ViT particularly suitable for grounded reporting, where the model must connect radiological findings to localized visual evidence. Unlike ImageNet (Ridnik et al., 2021) pretrained ViTs, CLIP's visual encoder produces features that are better structured for downstream language conditioning and cross-modal reasoning.

During training, the vision encoder remains frozen to preserve its pretrained CLIP representations. All images (16-bit) are first loaded and normalized by converting them to float in the range [0,1], applying percentile normalization (99th percentile), clipping, and scaling to 8-bit. The resulting grayscale image is replicated into three channels to form an RGB image compatible with the vision encoder's input requirements. Standard image transformations are then applied to produce the input tensor for ViT-B/32. The encoder outputs a 512-dimensional embedding, which is L2-normalized before being used as input to the language component.

### 3.5.2. LANGUAGE COMPONENT

For the language component, the pretrained GPT-2 model is used as the decoder. The visual embedding from the vision encoder is projected into the GPT-2 embedding space through a learnable mapping network, producing a sequence of prefix tokens that serve as conditioning context. The GPT-2 model receives these prefix embeddings concatenated with the token embeddings of the target finding text. During training the GPT-2 decoder is kept frozen; only the mapping network receives gradient updates. The model minimizes the next-token cross-entropy loss, ignoring padding tokens.

### 3.5.3. GROUNDING REPRESENTATION

Each Bounding box is represented using normalized coordinates $[x_1, y_1, x_2, y_2]$ within the range $[0, 1]$, defined relative to the image width and height. When rasterizing these bounding boxes, the normalized values are converted to pixel coordinates by scaling them to the image size, rounding to the nearest integer, and ensuring they fall within the image boundaries.

The Region image input $M \in \{0, 255\}^{H \times W}$ is a grounding mask, stencil, not a crop: all pixels inside any bounding box are set to 255 (white), and all others to 0 (black). The single channel is then replicated to RGB to match the vision encoder's expected 3-channel input.

The grounding mask $M$ is passed through the same preprocessing pipeline as the original image $I$. This preserves spatial alignment under the input transform: the grounding mask undergoes the same transforms, ensuring that its embedding corresponds to the same geometric regions emphasized in the image representation.

## 3.6. PadChest-GR Dataset

PadChest-GR (de Castro et al., 2025) (Grounded-Reporting), derived from the original PadChest dataset (Bustos et al., 2020), is designed to support the training of Grounded Radiology Report Generation models for chest X-ray (CXR) interpretation. It consists of a publicly available bilingual dataset containing 4,555 CXR studies with grounded reports (3,099 abnormal and 1,456 normal). Each study includes complete sets of sentences describing individual present (positive) and absent (negative) findings in both English and

Spanish. For this work, we rely exclusively on the English findings. In total, PadChest-GR provides 7,037 positive finding sentences and 3,422 negative ones. Each positive finding sentence is paired with up to two independent sets of bounding boxes annotated by different radiologists, along with categorical labels indicating the type of finding, anatomical region, and temporal progression. With its detailed localization information and rich annotations covering all clinically important findings, PadChest-GR constitutes a valuable dataset for developing and benchmarking GRRG models for CXR images.

For the training set, PadChest-GR comprises 3,185 studies, but since each study contains multiple findings, the data is divided into 7,315 individual entries, each corresponding to a specific image–finding pair. Similarly, the test set includes 915 studies, which, when divided per finding, result in 2,112 entries. This fine-grained structure enables models to learn more precise visual–textual alignments, improving their ability to associate image regions with specific radiological concepts.

## 4. Experiments and Results

Three different studies were conducted, each including multiple experiments, to thoroughly assess the improvement brought by adding grounding to the model. Each experiment was trained for 20 epochs using the AdamW (Loshchilov and Hutter, 2019) optimizer with a learning rate of 2e-5 and a linear warm-up of 500 steps. The batch size was set to 40, meaning that 40 independent samples, comprising an image, a segmentation mask, or both are processed in parallel during training.

For each individual sample, the visual embedding is mapped to a fixed sequence of 40 prefix tokens, which serve as conditioning context for GPT-2 before text generation begins. Thus, the number of prefix tokens per sample is 40 and is independent of the batch size.

For generation, we used beam search decoding with a beam size of 5, generation length of 15 tokens, and temperature set to 1.0. The best checkpoint was selected based on the lowest validation loss.

Our models are evaluated with two metric families. Among NLG metrics, BLEU (Papineni et al., 2002), ROUGE-L (Lin, 2004), and CIDEr (Vedantam et al., 2015) capture surface-level overlap and coverage of the reference text, while BERTScore (Zhang et al., 2020) provides a deeper semantic similarity measure. For clinical correctness, F1CheXbert (Smit et al., 2020) reflects agreement with clinical labels, and F1RadGraph (Jain et al., 2021) evaluates entity- and relation-level factual accuracy. Together, these metrics assess both linguistic quality and medical fidelity.

### 4.1. Study 1: Adding Grounding

The first study compared three configurations: using only the chest X-ray images, using the grounding masks alongside the chest X-ray images, and using only the grounding masks as input. This setup aimed to evaluate how explicit anatomical grounding affects finding generation and factual consistency.

As observed in Table 1 adding anatomical masks as a grounding signal significantly improved every metric compared to the baseline with only images. The model trained with images and grounding masks achieved the highest performance across all metrics, with especially strong gains in CIDEr (+42.38), BERTScore (+15.41), and CheXbert (+17.79),

Table 1: Comparison of model performance using only images, images and grounding masks, and only grounding mask. The best results are shown in **bold**.

| Metric | Only Images | Images and Grounding Masks | | Only Grounding Masks | |
|---|---|---|---|---|---|
| | | Value | Δ | Value | Δ |
| BLEU-1 | 9.11 | **18.60** | +9.49 | 15.16 | +6.05 |
| BLEU-2 | 4.81 | **12.96** | +8.15 | 10.04 | +5.23 |
| BLEU-3 | 3.00 | **9.12** | +6.12 | 6.87 | +3.87 |
| BLEU-4 | 2.09 | **6.40** | +4.31 | 4.74 | +2.65 |
| ROUGE-L | 10.79 | **24.11** | +13.32 | 20.97 | +10.18 |
| CIDEr | 16.36 | **58.75** | +42.38 | 43.24 | +26.88 |
| BERTScore | 24.02 | **39.43** | +15.41 | 33.33 | +9.31 |
| F1CheXbert | 7.68 | **25.46** | +17.79 | 21.43 | +13.75 |
| F1RadGraph | 1.31 | **9.31** | +7.99 | 6.56 | +5.25 |

confirming that grounding improves both linguistic and factual accuracy. The only grounding masks configuration still outperformed the only images baseline, demonstrating that structural cues alone are informative, but performance slightly decreased without the raw image, suggesting that both modalities are complementary.

## 4.2. Study 2: A Percent-Scaled Analysis

Since the introduction of grounding masks provided consistent benefits, we performed a second study that incrementally increased the percentage of the train set used. Two configurations were evaluated: using only the grounding masks, and using both the grounding masks and the chest X-ray images. This experiment aimed to identify how much grounding is optimal before reaching performance saturation.

Table 2: Performance comparison across different percentage of the train set using only grounding masks, and images and grounding masks. The best results are shown in **bold**.

| Experiment | BLEU-1 | BLEU-2 | BLEU-3 | BLEU-4 | ROUGE-L | CIDEr | BERTScore | F1CheXbert | F1RadGraph |
|---|---|---|---|---|---|---|---|---|---|
| 5% Only Grounding Masks | 9.58 | 4.96 | 1.96 | 0.48 | 11.02 | 5.58 | 18.89 | 5.17 | 0.63 |
| 5% Images and Grounding Masks | 7.29 | 3.72 | 0.80 | 0.00 | 10.00 | 2.05 | 10.69 | 5.14 | 0.44 |
| 10% Only Grounding Masks | 11.35 | 6.27 | 2.72 | 0.00 | 16.80 | 10.39 | 27.20 | 6.24 | 0.54 |
| 10% Images and Grounding Masks | 14.08 | 8.21 | 4.40 | 0.00 | 20.17 | 20.42 | 26.70 | 6.19 | 6.67 |
| 20% Only Grounding Masks | 16.10 | 9.48 | 5.43 | 3.27 | 20.63 | 31.45 | 34.33 | 10.82 | 4.43 |
| 20% Images and Grounding Masks | 16.75 | 10.26 | 5.99 | 3.68 | 21.82 | 37.70 | 33.00 | 14.46 | 4.24 |
| 50% Only Grounding Masks | 17.90 | 11.51 | 7.59 | 4.98 | 23.49 | 47.39 | 38.46 | 20.83 | 7.93 |
| 50% Images and Grounding Masks | 18.16 | 12.41 | 8.63 | 6.05 | 23.37 | 52.65 | 37.73 | 22.64 | 7.13 |
| 80% Only Grounding Masks | 13.44 | 9.65 | 6.77 | 4.75 | 18.65 | 45.73 | 35.58 | 21.07 | 7.74 |
| 80% Images and Grounding Masks | 17.92 | 12.57 | 8.99 | **6.50** | 23.89 | 56.39 | 39.23 | 24.46 | 8.62 |
| 100% Only Grounding Masks | 15.16 | 10.04 | 6.87 | 4.74 | 20.97 | 43.24 | 33.33 | 21.43 | 6.56 |
| 100% Images and Grounding Masks | **18.60** | **12.96** | **9.12** | 6.40 | **24.11** | **58.75** | **39.43** | **25.46** | **9.31** |

As observed in Table 2 incrementally increasing the masking ratio produced consistent performance gains across most metrics. Taking F1CheXbert as the primary indicator of clinical factual accuracy, Figure 2 shows a clear trend: both the only grounding mask and the image and grounding mask configurations surpass the 100% only image baseline as early as 20% of the training set. This result has practical implications for dataset creation: to match or exceed the performance of a model trained exclusively on full-image data, it is

not necessary to annotate the entire dataset. Instead, annotating approximately 20% of the chest X-ray scans with bounding boxes is sufficient to reach and even surpass the full-image baseline. This substantially reduces the annotation burden for radiologists while still achieving superior factual and semantic accuracy.

The highest overall performance was achieved using the complete images and grounding masks configuration, reinforcing that integrating spatial attention from segmented regions enhances both textual fluency and clinical accuracy.

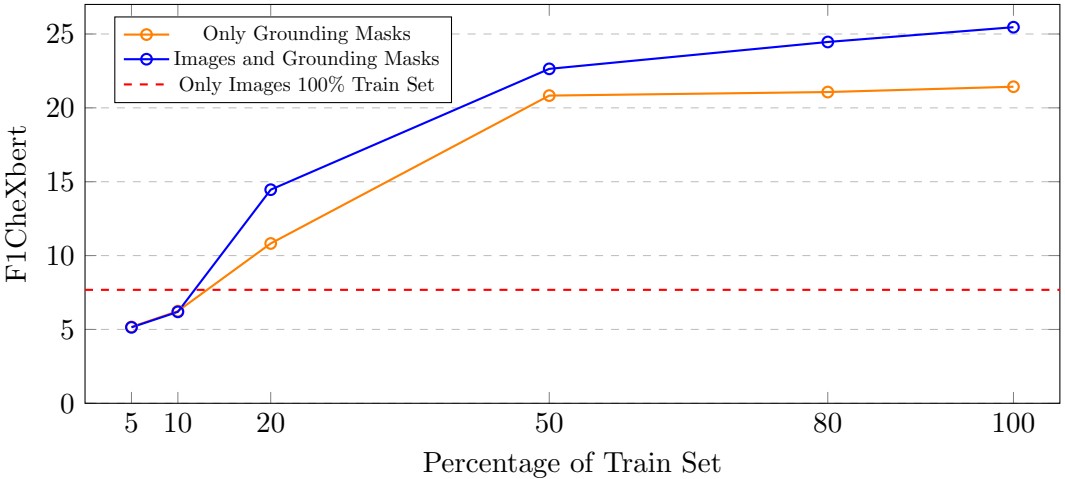

Figure 2: F1CheXbert comparison across training percentages.

### 4.3. Study 3: Bounding Box Noise Sensitivity Analysis

We introduce controlled noise into the ground-truth bounding boxes by applying random translation and isotropic scaling in normalized image coordinates. Each box is translated by up to $\pm16\%$ of the image size and scaled by a factor of $1 \pm 50\%$, with both distortions modulated by a level-dependent factor $l/5$, where $l \in \{1, \ldots, 5\}$. Bounding boxes are clipped to image boundaries, and invalid boxes are reverted to the original annotations.

Table 3: Performance comparison across different noise levels. The best results are shown in **bold**.

| Experiment | Noise | | BLEU-1 | BLEU-2 | BLEU-3 | BLEU-4 | ROUGE-L | CIDEr | BERTScore | F1CheXbert | F1RadGraph |
| | Translation | Scaling | | | | | | | | | |
|---|---|---|---|---|---|---|---|---|---|---|---|
| No Noise (L0) | 0% | 0% | **18.60** | 12.96 | 9.12 | 6.40 | 24.11 | 58.75 | 39.43 | 25.46 | 9.31 |
| Noise Level 1 (L1) | ±3.2% | ±10% | 18.54 | **13.14** | **9.38** | **6.63** | **25.01** | **62.32** | **40.16** | 25.57 | **10.02** |
| Noise Level 2 (L2) | ±6.4% | ±20% | 18.42 | 12.91 | 9.15 | 6.53 | 24.07 | 59.01 | 39.30 | **26.27** | 9.52 |
| Noise Level 3 (L3) | ±9.6% | ±30% | 17.40 | 12.43 | 9.02 | 6.57 | 23.38 | 60.86 | 39.23 | 24.43 | 9.68 |
| Noise Level 4 (L4) | ±12.8% | ±40% | 16.32 | 11.55 | 8.31 | 5.90 | 23.12 | 58.90 | 39.14 | 22.68 | 9.67 |
| Noise Level 5 (L5) | ±16% | ±50% | 16.00 | 11.16 | 7.96 | 5.68 | 21.74 | 56.75 | 38.00 | 22.39 | 9.31 |

As shown in Table 3, increasing bounding-box noise generally leads to a degradation in performance, reflecting the impact of spatial noise on grounding quality. Nevertheless, even under the highest noise level (L5), which severely perturbs the ground-truth boxes, the

model still outperforms the image-only baseline, indicating that highly coarse localization cues remain beneficial. Interestingly, low noise levels (L1 and L2) often improve performance across several metrics, suggesting a data-augmentation effect that promotes more robust visual–textual alignment.

Importantly, these results demonstrate that the effectiveness of the proposed method does not critically depend on perfectly precise or exhaustive bounding box annotations: even sparse, partial, or coarse grounding signals provide sufficient supervision to guide the generation process. This indicates that implicit or weakly supervised grounding can already yield meaningful benefits, supporting the robustness and practical applicability of the proposed approach.

### 4.4. Explainability

To analyze the effect of our grounding masks, we employ an attention-based explainability approach derived from the RegionFormer module. Specifically, we extract the self-attention maps between the learned prefix tokens and the pooled CLIP image embeddings across multiple Transformer layers. These attention maps are averaged over attention heads and layers and projected back onto the image grid, providing a qualitative indication of where the model attends when conditioning text generation. While these visualizations do not yield precise localization of individual findings, partly due to the limited size of the training dataset, they enable a consistent analysis of how grounding information influences the model's focus. When grounding masks are included, the model places greater emphasis on the corresponding pathological regions, as illustrated in the attention maps in Figure 3. Moreover, the grounding masks not only highlight the relevant abnormalities but also guide the model on where to focus. This effect is evident in the final example, where the model generates new attention around the pacemaker region, even though it was not originally emphasized. This indicates that grounding masks not only boost finding generation accuracy but also improve interpretability by guiding the model to focus on clinically meaningful areas.

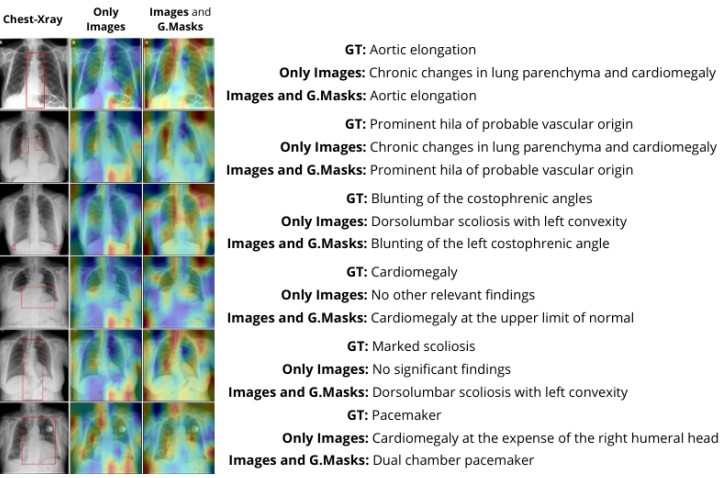

Figure 3: CXR images with attention maps using only image vs image and grounding mask.

## 5. Conclusion

This work shows that incorporating spatial grounding substantially enhances Radiology Report Generation by guiding models toward clinically relevant visual evidence. Although performance is limited by the small size of the PadChest-GR dataset (4,555 studies), grounding masks still yield clear gains in both linguistic quality and clinical accuracy.

By combining ViT embeddings with a transformer mapping module and leveraging grounding masks derived from annotated bounding boxes, our framework enables the decoder to generate more accurate, specific, and interpretable reports.

Experiments on PadChest-GR show consistent improvements across all metrics when using grounding. The configuration that only uses the grounding masks performs surprisingly well compared to the image-only configuration, which often struggles to align findings with text and produces unfocused descriptions. Grounding masks alone provide strong cues about the expected nature of the finding, even though this output is not derived from image features, as the model does not actually see the underlying image. Despite this, the grounding-based cues alone still perform substantially better than the image-only configuration. When image and mask are combined, the model benefits from both visual context and spatial guidance, achieving the best overall performance.

An important outcome of the percent-scaled analysis is that grounding remains highly effective even when bounding box annotations are available for only a fraction of the training data. Our results show that incorporating grounding information for approximately 20% of the training set is sufficient to match or surpass the performance of a model trained exclusively on full-image data across both linguistic and clinical metrics. This demonstrates that exhaustive annotation of the entire dataset is not required to obtain substantial gains, significantly reducing the annotation burden for radiologists.

The noise sensitivity analysis further demonstrates the robustness of the proposed grounding approach to imprecise spatial supervision. Even under severe perturbations of the bounding box annotations, the model consistently outperforms the image-only baseline, indicating that highly coarse localization cues remain beneficial for guiding report generation. Moreover, low levels of noise can even improve performance, suggesting a data-augmentation effect that promotes more robust visual–textual alignment. These findings confirm that the proposed method does not critically depend on perfectly precise bounding box annotations, and that sparse, noisy, or weakly supervised grounding signals are sufficient to provide meaningful benefits in practice.

While this study focuses on isolating the effect of explicit spatial grounding, several natural extensions remain. An important direction is to combine the proposed grounding mechanism with domain-specific vision encoders pretrained on large-scale medical imaging data, as well as alternative backbone architectures, to further assess potential performance gains. In addition, the grounding strategy is largely decoder-agnostic and can be readily integrated with larger or medical-domain language models. We expect that pairing explicit spatial grounding with more powerful or clinically adapted LLMs may further improve factual accuracy and reduce hallucinations, an avenue we plan to explore in future work.

Overall, our results indicate that grounding is a highly beneficial component for reliable and interpretable radiology report generation, with consistent gains observed even under weak or reduced spatial supervision.

## Acknowledgments

This work was supported by the Generalitat Valenciana under the grant CIPROM/2023/17.

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

## Appendix A. Forward Pass Implementation of the Transformer Mapper

---

**Algorithm 1** RegionFormer Pseudocode

---

```python
def forward(x):
    """
    Forward propagation of the RegionFormer Mapper.

    Args:
        x: Input visual embedding (Image, Mask, or Concatenation).
            Shape: (Batch_Size, Input_Dim)
    Returns:
        Refined prefix tokens for the language model.
            Shape: (Batch_Size, Prefix_Length, Hidden_Dim)
    """

    # 1. Projection & Reshape
    # Project the global 1D visual vector into a flattened sequence,
    # then reshape into latent visual tokens.
    # x: (Batch, Input_Dim) -> (Batch, Clip_Length, Hidden_Dim)
    x = self.linear(x).view(x.shape[0], self.clip_length, -1)

    # 2. Learnable Prefixes Expansion
    # Expand the fixed learnable parameters to match the batch dimension.
    # prefix: (Batch, Prefix_Length, Hidden_Dim)
    prefix = self.prefix_const.unsqueeze(0).expand(x.shape[0], *self.prefix_const.shape)

    # 3. Sequence Concatenation
    # Concatenate projected visual tokens (x) and learnable prefix tokens (prefix).
    # Resulting sequence contains [Visual_Tokens, Prefix_Tokens].
    # Shape: (Batch, Clip_Length + Prefix_Length, Hidden_Dim)
    combined_seq = torch.cat((x, prefix), dim=1)

    # 4. Self-Attention Refinement
    # Process the combined sequence. The learnable prefixes attend to the
    # visual tokens and to themselves via full self-attention.
    out = self.transformer(combined_seq)

    # 5. Output Extraction
    # Slice the output to discard the visual tokens, retaining only
    # the refined prefix embeddings to be used by the LLM.
    # Return Shape: (Batch, Prefix_Length, Hidden_Dim)
    return out[:, self.clip_length:]
```

---

