# OpenReview forum: "Does Grounding Improve Radiology Report Generation? An Empirical Study on PadChest-GR"
_MIDL.io/2026/Conference — MIDL 2026 Poster_

### Official Review · Reviewer_ustC · 2025-12-31

**Confidence:** 4
**Preliminary Rating:** 3
**Final Rating:** 4

**Summary:**

This paper addresses a central challenge in Radiology Report Generation (RRG): the lack of explicit grounding between visual evidence and generated textual findings, which often leads to incomplete descriptions, generic phrasing, and hallucinations. To mitigate this issue, the authors propose a grounding-based RRG framework that incorporates spatially localized visual cues in the form of bounding boxes for anatomical regions and pathologies. The framework connects a ViT-based vision encoder with a GPT-2 language decoder via a lightweight transformer-based bridging module inspired by bridge-enhanced VED architectures. In addition, a region-to-text generation task is introduced, encouraging the model to generate findings directly from localized regions of interest. Experiments on the PadChest-GR dataset demonstrate consistent improvements across both language-quality metrics and clinically meaningful evaluation metrics. Overall, the work tackles an important problem in medical vision–language modeling, though several assumptions regarding grounding annotations, model design choices, and robustness warrant further clarification and validation.

**Strengths:**

The paper targets a well-known limitation of RRG systems, namely hallucination and weak visual grounding.
Introducing explicit spatial grounding via bounding boxes is intuitive and clinically meaningful.
The proposed region-to-text task provides a clear mechanism to enforce localized visual–text alignment.

**Weaknesses:**

The method relies heavily on the availability and accuracy of grounding box annotations, which may limit scalability and real-world applicability.
The necessity of bounding-box supervision is not sufficiently discussed or justified relative to weaker or implicit grounding alternatives.

**Detailed Comments:**

Dependence on Grounding Box Annotations
The proposed framework assumes access to bounding boxes for anatomical regions and pathologies. However, such annotations are expensive and often unavailable in large-scale radiology datasets. The authors should discuss:
How the method would generalize to datasets without grounding box labels.
The impact of annotation noise or imprecision on model performance.
Whether the framework could be extended to weaker forms of supervision.
Why Not Learn Grounding Directly from Image–Report Pairs?
Given recent advances in weakly supervised and contrastive vision–language learning, it is unclear why the model does not attempt to learn grounding implicitly from image–report pairs. A discussion or comparison with phrase grounding or attention-based alignment approaches would help position the contribution more clearly.
Role of Learnable Embeddings in the Bridging Module
The paper introduces learnable embeddings in the bridging transformer, but their functional role is not sufficiently explained. The authors should clarify:
What information these embeddings encode.
Whether they model region identity, modality alignment, or positional information.
How sensitive the model is to the design of these embeddings.
Choice of Language Model (GPT-2 vs. Larger LLMs)
The framework employs GPT-2 as the language decoder. It would be helpful to understand:
Why GPT-2 was chosen over more recent models such as LLaMA or domain-adapted medical LLMs.
Whether the grounding benefits persist when stronger language models are used.
If the grounding mechanism compensates for weaker language modeling capacity.

**Justification Of Final Rating:**

The paper is well motivated and technically solid, with clear contributions to medical grounding. The authors’ response adequately addressed my concerns. Although additional comparisons with recent VLM-based methods could further strengthen the work, the overall quality and contribution justify a positive final rating.

**Justification Of The Preliminary Rating:**

This paper addresses an important and relevant problem in medical report generation by explicitly incorporating spatial grounding, and the proposed framework is well-motivated with encouraging experimental results. The use of clinically grounded evaluation metrics is a strong point. However, the reliance on high-quality grounding annotations, limited discussion of alternative grounding strategies, and lack of robustness analysis raise concerns regarding scalability and generalization. With clearer motivation, additional ablations, and robustness studies, the paper could become a strong contribution to the MIDL community.

**Questions To Address In The Rebuttal:**

How critical are precise bounding box annotations to the proposed method’s success?
Can the authors provide ablation studies comparing explicit grounding with implicit or weakly supervised grounding?
Would the proposed grounding strategy still be effective when paired with larger or medical-domain LLMs?

---

> ### Author Response · Authors · 2026-01-25
>
> The authors sincerely thank the reviewer for the thoughtful and constructive feedback. We have carefully revised the manuscript in response to these comments, resulting in significant improvements in clarity, structure, and content. Below, we address each point in detail.
>
> ---
> - **Sensitivity to bounding box precision and role of weak grounding.**
>   We thank the reviewer for this important question. A key takeaway of our work is that the proposed approach does not rely on perfectly precise or exhaustive bounding box annotations. In addition to the percent-scaled analysis (Study 2), which demonstrates that full grounding coverage is not required to achieve competitive or improved performance, we introduced a new ablation study explicitly analyzing robustness to imprecise grounding. This new study shows that the model remains effective even when bounding boxes are substantially perturbed, indicating that coarse, partial, or weak localization cues are sufficient to guide the generation process. Together, these results support the practicality of the method in scenarios where grounding is sparse, noisy, or automatically generated.
>   - *Change made:* We added *Study 3: Bounding Box Noise Sensitivity Analysis*.
>   - *Location:* Section 4.3.
> ---
> - **Compatibility with larger or medical-domain language models.**
>
>   We thank the reviewer for raising this point. The proposed grounding strategy is largely *model-agnostic* with respect to the language decoder, as grounding is injected through learned visual prefix tokens produced by the RegionFormer, rather than through assumptions specific to GPT-2.
>
>   As a result, the same grounding mechanism can be paired with larger or medical-domain LLMs, such as clinically adapted Transformers, without architectural changes. In fact, we expect grounding to become even more beneficial in such settings, as larger or domain-specific language models typically have stronger generative capacity but are also more prone to hallucinations when visual evidence is weakly constrained. Explicit spatial grounding can act as an additional inductive bias that anchors generation to relevant image regions, potentially improving factual accuracy and interpretability.
>
>   While evaluating larger or medical-domain LLMs is beyond the scope of this study, our framework is designed to support such extensions seamlessly, and we consider this an important direction for future work.
>   - *Change made:* We added it as future work in the conclusions.
>   - *Location:* Section 5.
> ---
> - **Following the reviewer’s recommendation, we clarified the functional role of the learnable embeddings in the bridging transformer. In particular, we expanded the explanation in the last paragraph of Section 3.2, explicitly describing the type of information these embeddings encode and their role in modeling region identity and modality alignment within the bridging module.**
> ---
> We once again thank the reviewer for the invaluable feedback and hope that the revisions have adequately addressed all concerns.

---

> > ### Comment · Reviewer_ustC · 2026-02-02
> >
> > Thanks for the authors’ detailed response. I think my concerns have been addressed very well. Nevertheless, I note that several recent VLM-related works on medical phrase grounding and explainable medical VQA are not included in the current comparison. I therefore suggest that the authors consider citing the following representative papers and clarifying their relationship to the proposed method:
> > Medical phrase grounding with region-phrase context contrastive alignment
> > Uncertainty-aware medical diagnostic phrase identification and grounding
> > GEMeX: A large-scale, groundable, and explainable medical VQA benchmark for chest X-ray diagnosis

---

> > > ### Author Response · Authors · 2026-02-03
> > >
> > > Thank you for the positive assessment and for confirming that your concerns have been satisfactorily addressed. We also appreciate you pointing out these relevant and recent VLM-related works on medical phrase grounding and explainable medical VQA.
> > >
> > > We agree that the suggested papers are closely related to our research direction. In the revised manuscript, we will cite these representative works and include a discussion clarifying their relationship to our proposed method. In particular, we will highlight the differences in task formulation, supervision signals, and model objectives, as well as how our approach complements existing efforts in phrase grounding and explainable medical VQA.
> > >
> > > Thank you again for this constructive suggestion, which we believe will further strengthen the contextualization of our work within the current literature.

---

### Official Review · Reviewer_5NXS · 2026-01-09

**Confidence:** 4
**Preliminary Rating:** 3
**Final Rating:** 4

**Summary:**

The paper presents a method for grounded radiology report generation to mitigate issues of existing approaches such as hallucinations or incomplete findings generation. The proposed system leverages frozen vision and language models that are connected via a learnable "bridge". Experiments on the PadChest-GR dataset show that incorporating both images and grounding masks consistently improves lexical and clinical evaluation metrics. Moreover, the proposed image+mask model outperforms its image-only counterpart using only a subset (20%) of the original training set, thus significantly reducing the associated annotation cost.

**Strengths:**

- The grounded radiology report generation approach is clearly motivated in the Introduction section.

- The proposed methodology is thoroughly described and fairly easy to follow.

- Source code is provided on a public Github repository.

- Overall, this is an interesting method that represents a promising step towards interpretable radiology report generation models and would be of value to the community.

**Weaknesses:**

- There are some parts of the main text that are slightly confusing. For example, Figure 1 (pg. 4) shows that a sequence of visual patch embeddings is extracted from the vision encoder (ViT) and passed to the RegionFormer, whereas in Sections 3.5.1 and 3.5.2 (pg. 6) the authors refer to "a single visual vector $v$" in the main text.

- The visualizations depicted in Figure 3 (pg. 10) are not convincing. It seems that the attention maps for the **Images and G. Masks** setup also activate over the entire image.

- The requirement for "perfect" grounding masks is too strict, since such masks would not be available in most real-world cases.

**Detailed Comments:**

- The arrangement of sections can be improved: for example, Sections 3.1-3.3 could be merged with the overview of the framework presented in Section 3.5.

- Is there a connection between structured radiology report generation [1] and grounded report generation? If so, it would be useful to explain this in the paper.

- It would also be useful to discuss the differences between your proposed method and other approaches such as MAIRA-Seg [2].



[1] Delbrouck, Jean-Benoit, et al. "Automated structured radiology report generation." Proceedings of the 63rd Annual Meeting of the Association for Computational Linguistics (Volume 1: Long Papers). 2025.

[2] Sharma, Harshita, et al. "Maira-seg: Enhancing radiology report generation with segmentation-aware multimodal large language models." arXiv preprint arXiv:2411.11362 (2024).

**Justification Of Final Rating:**

The paper has been improved during the rebuttal period and now includes an additional experiment showing that the proposed method is reasonably robust to noisy bounding box supervision, which addresses the main concern in my initial review.

That being said, some of the design choices related to the model architecture are suboptimal and limit its off-the-shelf (i.e., image-only) performance. More specifically, the choice of pre-trained CLIP vision encoder, which is not adapted to the chest X-ray modality, and the use of pooled CLIP embeddings instead of patch-level image features may restrict the model's ability to capture fine-grained spatial features.

Despite these limitations, the proposed grounded report generation method is well-motivated and model-agnostic. I believe that this aspect of the work, given the growing interest of the medical imaging community in grounding and interpretability, is a solid contribution on its own.

Therefore, I am willing to increase my score to 4 (Weak accept).

**Justification Of The Preliminary Rating:**

The method introduced in this paper shows merit and would benefit from minor revisions to the main text and some additional experiments/ablations. Also, relaxing the requirement for "perfect" pathology grounding masks would improve its applicability to real-world scenarios.

**Questions To Address In The Rebuttal:**

- From what I understand from the main text, the authors used a vision encoder (CLIP ViT) trained on RGB images for their experiments. How about using a publicly available CLIP ViT trained on chest X-rays? Also, what about other vision encoder architectures (e.g., DenseNet)?

- Would it be useful to pass cropped image features, instead of image+mask features, as input to the RegionFormer?

- Instead of ground truth pathology bounding boxes, can the model use either anatomical bounding boxes or other silver standard annotations (e.g., outputs of a pathology detector or a phrase grounding model)?

---

> ### Author Response · Authors · 2026-01-25
>
> The authors sincerely thank the reviewer for the thoughtful and constructive feedback. We have carefully revised the manuscript in response to these comments, resulting in significant improvements in clarity, structure, and content. Below, we address each point in detail.
>
> ---
> - **Choice of vision encoder and potential alternatives.**
>
>   We thank the reviewer for this insightful suggestion. We agree that employing a CLIP ViT model pretrained specifically on chest X-rays, or other large-scale medical imaging corpora, would be a very interesting direction and could potentially lead to further performance improvements. Similarly, exploring alternative vision encoder architectures such as DenseNet-based models, is a promising avenue.
>
>   That said, the primary objective of this work was not to optimize the vision backbone itself, but rather to study *whether and to what extent explicit spatial grounding improves radiology report generation*. In this context, we deliberately fixed the vision encoder to isolate the effect of adding grounding information via bounding boxes. Our experimental results show that, even with a generic CLIP ViT trained on natural images, the introduction of bounding-box-based grounding consistently improves both linguistic quality and clinical correctness.
>
>   We believe this finding is important, as it suggests that grounding is beneficial independently of the specific encoder choice. Nonetheless, we fully agree that combining grounding with domain-specific encoders or alternative architectures is a natural and valuable extension, and we plan to explore these directions in future work.
>   - *Change made:* We added it as future work in the conclusions.
>   - *Location:* Section 5.
> ---
> - **Use of cropped image features versus image+mask features.**
>
>   We thank the reviewer for this thoughtful question. Indeed, using cropped image features extracted directly from the bounding boxes is a valid and interesting alternative that we carefully considered. However, in this work we opted to use the combination of the full image and a grounding mask in order to encourage the model to attend to both *local* and *global* visual context.
>
>   This design choice was motivated in part by the characteristics of the PadChest-GR dataset, where some findings are associated with multiple bounding boxes and where global anatomical context can be important for correct interpretation. By preserving the full spatial layout of the image, the model can reason about the relative position of regions and maintain awareness of global structures, rather than relying solely on isolated crops. Our results suggest that this balance between global context and localized emphasis is beneficial for grounded report generation.
> ---
> - **Use of alternative or weaker forms of grounding supervision.**
>   We agree that alternative grounding signals, such as anatomical bounding boxes, silver-standard annotations from pathology detectors, or outputs of phrase-grounding models, are both feasible and highly relevant. A central motivation of our work is to demonstrate that *weak grounding* can still effectively guide the generation process. Beyond the percent-scaled experiments, which already suggest that full grounding coverage is not required to achieve competitive or improved performance, we added a new ablation study analyzing robustness to noisy bounding boxes. This study shows that the model remains effective under substantial spatial imprecision, indicating that coarse or imprecise localization cues still provide meaningful supervision and supporting the applicability of the method with weak or automatically generated grounding signals.
>   - *Change made:* We added *Study 3: Bounding Box Noise Sensitivity Analysis*, which evaluates model robustness under increasing levels of bounding box noise.
>   - *Location:* Section 4.3.
> ---
> - **Following the reviewer’s recommendations, we improved the explanation of the explainability component in Section 4.4, added a comparison between Structured RRG and Grounded RRG in the final paragraph of Section 2 (Related Work), and incorporated the MAIRA-Seg model [2] into the discussion of existing approaches in the first and second paragraphs of the Related Work section. We also clarified that RegionFormer relies on a single pooled CLIP visual embedding rather than ViT patch-level tokens, revised Figure 1 accordingly, and updated the main text to ensure consistency between the figure and the actual implementation. Finally, we reordered the sections of the manuscript in accordance with the first reviewer’s recommendations.**
> ---
> We once again thank the reviewer for the invaluable feedback and hope that the revisions have adequately addressed all concerns.

---

### Official Review · Reviewer_ijLA · 2026-01-10

**Confidence:** 5
**Preliminary Rating:** 2

**Summary:**

This paper introduces a lightweight transformer-based bridging module for grounded radiology report generation, i.e. to generate a medical report for a specified region of interest (given to the model by a grounding mask obtained from a bounding-box annotation). The proposed model is tested on PadChest-GR and experiments show that the best results are obtained when both image and grounding masks are used.

**Strengths:**

The proposed model is lightweight, as it keeps both the vision encoder and text decoder frozen.

The experimental setup is well designed, as it compares the model using only images, only grounding masks, and both.

**Weaknesses:**

The authors claim that they introduce a novel task "Region-to-Text". However, this task is not novel [1-5].

The proposed method is not compared against any other methods.

Section 4.3., in its current form, serves no purpose, since it only shows qualitative results, which do not allow concluding that including the grounding masks improves explainability. Qualitative results with explainability metrics are needed.

[1] Rubel, Al Shahriar, Frank Shih, and Fadi Deek. "RoI-MedCap: Region of Interest-Based Medical Image Captioning with Multi-Stream Connector." IEEE-EMBS International Conference on Biomedical and Health Informatics 2025.

[2] Rubel, Al Shahriar, Frank Y. Shih, and Fadi P. Deek. "MoE-MSC: Mixture of Experts with Multi-Stream Connector for Modality-Aware Medical Image Captioning." Proceedings of the 16th ACM International Conference on Bioinformatics, Computational Biology, and Health Informatics. 2025.

[3] Luo, Lingxiao, et al. "Vividmed: Vision language model with versatile visual grounding for medicine." Proceedings of the 2025 Conference of the Nations of the Americas Chapter of the Association for Computational Linguistics: Human Language Technologies (Volume 1: Long Papers). 2025.

(not peer reviewed)
[4] Xiao, Xi, et al. "Describe Anything in Medical Images." arXiv preprint arXiv:2505.05804 (2025).

[5] Bannur, Shruthi, et al. "Maira-2: Grounded radiology report generation." arXiv preprint arXiv:2406.04449 (2024).

**Detailed Comments:**

Main comments:

The order of the subsections of Section 3 is confusing and could be improved. My suggestion would be to start with the general framework without specifying the vision encoder and text decoder used. So, I would suggest to start with what is now Section 3.5., then 3.5.1., 3.5.2 (merging it with 3.3.), 3.5.3, 3.5.4. Then would come a new section regarding "Implementation Details", where I would include 3.1., 3.2., and 3.3.1. Finally the dataset section. In summary:

3. Methodology

3.1. Visual Encoding (former 3.5.1.)

3.2. Visual-Language Mapping (former 3.5.2. + former 3.3.)

3.3. Language Generation (former 3.5.3.)

3.4. Grounded Generation Objective (former 3.5.4)

3.5. Implementation Details

3.5.1. Vision Component (former 3.1.)

3.5.2. Language Component (former 3.2.)

3.5.3. Grounding Representation (former 3.3.1.)

3.6. PadChest-GR Dataset (former 3.4.)

Minor comments:

The results of text generation metrics (Table 1) are usually presented multiplied by 100. It would also improve readability of the results.

"chest x-ray" sometimes is written as "chest x-ray". Please make it consistent throughout the document.

The title of Section 3 would be more informative if it just read "Methodology".

In Section 3.5. "Grounded Radiology Report Generation" has already been previously defined as GRRG.

The title of Section 4, where it reads "Experimentation" should read "Experiments". At the beginning of Section 4 where it reads "launched" should read "conducted".

In Section 4 there is a space missing before every reference to the metrics (e.g. after ROUGE-L, etc).

At the beginning of page 9 where it reads "Incrementally" should read "incrementally".

**Justification Of The Preliminary Rating:**

The authors claim to have proposed a novel task, which is not true (see the references included in the Weaknesses section). There is no comparison of the proposed method against other methods. However, the method is valid, so I am open to increasing my score if comparison against other methods is included.

**Questions To Address In The Rebuttal:**

See the Weaknesses comment regarding novelty and comparison to other methods.

In Section 3.5.2. the authors mention that "CLIP-ViT produces a single 512-dimensional embedding" and that "GPT-2 operates over a sequence of h-dimensional token embeddings". Both statements are not true. CLIP-ViT being a transformer architecture has, before pooling/CLS token, several embeddings and, in theory, all could be used as input to RegionFormer. GPT-2 can receive a single prefix token. Actually, afterwards in the text (see next comment) the authors mention 40 prefix embeddings. The statement is also contradicted by Figure 1, where there are several "Image embeddings", several "G.Mask embeddings", and several "Learnable embeddings".

The authors mention a batch size of 40, saying that "the model generates 40 prefix embeddings". How so? Are 40 images/masks processed simultaneously, each with 1 prefix token? Or a single image has 40 prefix tokens? How many prefix tokens are used as input to the language decoder? And if only one is being used per image, why not use more than one and compare the results?

Which explainability method is used to produce the maps of Section 4.3.?

---

> ### Author Response · Authors · 2026-01-25
>
> The authors sincerely thank the reviewer for the thoughtful and constructive feedback. We have carefully revised the manuscript in response to these comments, resulting in significant improvements in clarity, structure, and content. Below, we address each point in detail.
>
> ---
> - **Novelty and comparison to related methods.**
>   We thank the reviewer for the additional citations and for highlighting closely related works. While highly relevant, most of these works focus on related but distinct problem settings. One of the cited papers, *VividMed: Vision language model with versatile visual grounding for medicine*, focuses on generating bounding boxes as outputs rather than using bounding boxes as an explicit input to condition text generation, and therefore does not directly align with our region-to-text setting. As already discussed in the second paragraph of Related Work, other approaches such as RoI-MedCap and MoE-MSC visually embed bounding boxes into the image pixels (with code not yet publicly available), whereas our method treats bounding boxes as a separate structured input. The most closely related work is *Describe Anything in Medical Images*, which also models regions independently from the image using focal crops; we found this approach particularly interesting, but note that the lack of released code and the limited rebuttal timeframe prevented a direct empirical comparison.
>
>   Nevertheless, we found these contributions really valuable and have included them in the Related Work section.
>
>   We acknowledge that the region-to-text task itself is not novel, and therefore we have refocused the paper on grounding for radiology report generation. While our initial motivation was that prior works citing PadChest-GR did not explicitly address region-to-text generation, our primary contribution, as clarified in the third paragraph of the Introduction, is to rigorously evaluate the practical impact of grounding. Through ablation studies, we analyze whether conditioning generation on bounding boxes corresponding to anatomical regions and pathologies improves factual accuracy, clinical relevance, and descriptive quality.
>   - *Changes made:* Refocused the paper on grounding for radiology report generation and clarified distinctions with prior region-based methods.
>   - *Location:* Introduction (Section 1, paragraph 3) and Related Work (Section 2, paragraphs 1 and 2).
> ---
> - **Clarification of visual embeddings, prefix tokens, and training configuration.**
>   Although CLIP-ViT internally computes patch-level embeddings before pooling, our method uses *only the final pooled CLIP embedding* as the visual representation. RegionFormer never receives patch tokens or intermediate ViT features; instead, each image (or image–mask pair) is represented by a single global embedding. This embedding is expanded by RegionFormer into a *fixed-length sequence of 40 learned prefix tokens*, which provide richer conditioning for GPT-2. These prefix tokens are purely learned, token-like representations and do not correspond to image patches or spatial regions. We further clarify that the prefix length (40 tokens per sample) is independent of the batch size (40 samples processed in parallel during training). To avoid ambiguity, Figure 1 was updated to explicitly show a single pooled CLIP embedding being mapped into prefix tokens before conditioning GPT-2.
>   - *Change made:* We revised the text in Sections 3.2 and 4 to clarify the use of pooled CLIP embeddings, the learned nature and role of multiple prefix tokens, and the distinction between prefix length and batch size. We updated Figure 1 to reflect this pipeline unambiguously and added pseudocode in Appendix A.
>   - *Location:* Section 3.2 (Visual–language mapping), Section 4 (Experiments and Results), Figure 1, Appendix A.
> ---
> - **Clarification of the explainability approach used in Section 4.4.**
>   We thank the reviewer for this comment. We clarify that the visualizations shown in Section 4.4 are based on an *attention-based explainability approach* derived from the RegionFormer module, and are intended as qualitative indicators rather than precise lesion localization. The attention maps are obtained by extracting and averaging cross-attention weights between the learned prefix tokens and the pooled CLIP image embedding across multiple Transformer layers.
>   - *Change made:* We revised Section 4.4 to explicitly describe the attention-based explainability method and to clarify its qualitative nature.
>   - *Location:* Section 4.4 (Explainability).
> ---
> - **Following the reviewer’s recommendations, we have reorganized Section 3 to improve clarity and logical flow, adopting the proposed subsection order and renaming it to Methodology. In addition, all minor comments have been carefully addressed.**
> ---
> We once again thank the reviewer for the invaluable feedback and hope that the revisions have adequately addressed all concerns.

---

### Author Rebuttal · Authors · 2026-01-25

**Rebuttal:**

The authors sincerely thank the reviewers for their thoughtful and constructive feedback, all of which has been invaluable in improving the manuscript. Below, we summarize the main updates.

- Reordered Section 3 to improve clarity and reduce potential confusion.
- Clarified the methodology in Section 3.2, ensured consistency with Figure 1, and added an appendix containing pseudocode for completeness.
- Clarified the explainability analysis in Section 4.4.
- Added a concluding paragraph in the Related Work section comparing Structured RRG and Grounded RRG. We further discuss a potential hybrid approach that combines both methodologies to leverage their respective strengths.
- Added a new paragraph in Section 2 (Related Work) discussing the differences between our approach and existing models.
- Introduced a new ablation study in Section 4.3: *Bounding Box Noise Sensitivity Analysis*.
- Updated the conclusion to emphasize that neither full grounding coverage nor perfectly precise annotations are required, supporting the practical applicability of the proposed method.

The attached zip file contains the final revised submission PDF, as well as an additional PDF with added and deleted changes clearly marked.

We once again thank the reviewers for their invaluable feedback and hope that the revisions have adequately addressed all concerns.

**Supporting Material:**

/attachment/ff4db4df0465899b53388b8fac002819b5340c71.zip

---

### Meta-Review · Area_Chair_M9Sm · 2026-02-10

**Recommendation:** Accept (Poster)
**Confidence:** 4

**Metareview:**

This paper proposes a grounded radiology report generation framework that connects a frozen CLIP-ViT and GPT‑2 via a lightweight bridge, conditioning on bounding‑box masks and showing that image+mask input on PadChest‑GR improves both linguistic and clinical metrics over image‑only and mask‑only variants. Reviewers value the clear motivation, detailed methodology, released code, and added ablations demonstrating robustness to partial and noisy grounding as well as reduced annotation requirements. Remaining concerns include the limited novelty of the region‑to‑text formulation, lack of direct comparison to prior grounded/region‑based RRG or captioning baselines, and reliance on bounding‑box supervision with a non–chest‑X‑ray CLIP backbone that may underuse spatial detail. Overall, the work is technically sound, the revisions address most clarity and robustness issues, and the grounded RRG framework is likely to be useful to the community despite these limitations.

---

### Decision · Program_Chairs · 2026-02-13

Accept (Poster)